# Absolute dating of the European Neolithic using the 5259 BC rapid $^{14}$C excursion

Andrej Maczkowski [1,2] ✉, Charlotte Pearson[3], John Francuz[1], Tryfon Giagkoulis [4], Sönke Szidat [2,5], Lukas Wacker [6], Matthias Bolliger [1,2,7], Kostas Kotsakis[4] & Albert Hafner [1,2]

Abrupt radiocarbon ($^{14}$C) excursions, or Miyake events, in sequences of radiocarbon measurements from calendar-dated tree-rings provide opportunities to assign absolute calendar dates to undated wood samples from contexts across history and prehistory. Here, we report a tree-ring and $^{14}$C-dating study of the Neolithic site of Dispilio, Northern Greece, a waterlogged archaeological site on Lake Kastoria. Findings secure an absolute, calendar-dated time using the 5259 BC Miyake event, with the final ring of the 303-year-long juniper tree-ring chronology dating to 5140 BC. While other sites have been absolutely dated to a calendar year through $^{14}$C-signature Miyake events, Dispilio is the first European Neolithic site of these and it provides a fixed, calendar-year anchor point for regional chronologies of the Neolithic.

The Neolithic period in western Eurasia marks one of the most important transitions in human social, economic, and technological history. This transition, which lasted several millennia, is chiefly characterised by the appearance and gradual adoption of agriculture and animal husbandry accompanied by increasing social and material culture complexity. The appearance of the first Neolithic communities on the Aegean coasts and in Northern Greece, and hence in continental Europe, is dated to around 6500 BC[1–6]. The expansion of the Neolithic settlements further inland to the North was a rapid process[7]. Consequently, their precise dating is essential to our understanding of neolithisation processes in Europe and critical to assessments of the environmental footprint of the new farming subsistence practices. However, the temporal resolution of archaeological and environmental proxies in the region is highly variable, producing significant discrepancies between various chronological and terminological systems that deal with the periodisation of the Neolithic[8]. The combination of tree-ring dating (dendrochronology) and rapid $^{14}$C excursions has been previously used for the absolute dating of sites only from the Common Era[9–12]. Here, we present the absolute dating of the 6th millennium BC lake-dwelling site of Dispilio in Northwestern Greece. Our

results may provide the basis for absolute dendrochronological dating of other Neolithic sites in the region (Fig. 1).

Tree-rings enable high-resolution dating, the possibility of annually resolved climatic reconstruction, and multidisciplinary chronological synchronization, often to a single growth season of a specific calendar year[13]. Until recently[9], dendrochronological dating was possible only against reference tree-ring chronologies, which are continuous, unbroken sequences of tree-ring width records extending from the present back to the distant past. Based on known dates from living trees, calendar-years are assigned to tree-rings and then extended into the past using climatically constrained, region-specific, continuous tree-ring growth patterns. Long-term concentrated efforts in search for old wood samples have resulted in the construction of long tree-ring records extending back many thousands of years; these are widely applied to dating[14–16] and paleoclimatic analyses[17,18], shedding light on past human and environmental interactions. However, these records are geographically limited and rare. Moreover, many prehistoric tree-ring chronologies are only approximately constrained on a calendar timescale through conventional $^{14}$C wiggle-matching and have no absolute calendar anchor.

[1]Institute of Archaeological Sciences, University of Bern, Bern, Switzerland. [2]Oeschger Centre for Climate Change Research, University of Bern, Bern, Switzerland. [3]Laboratory of Tree-Ring Research, University of Arizona, Tucson, USA. [4]School of History and Archaeology, University of Thessaloniki, Thessaloniki, Greece. [5]Department of Chemistry, Biochemistry and Pharmaceutical Sciences, University of Bern, Bern, Switzerland. [6]Laboratory for Ion Beam Physics, ETH Zürich, Switzerland. [7]Laboratory for Dendrochronology, Archaeological Service of Canton of Bern, Bern, Switzerland. ✉e-mail: andrej.maczkowski@unibe.ch

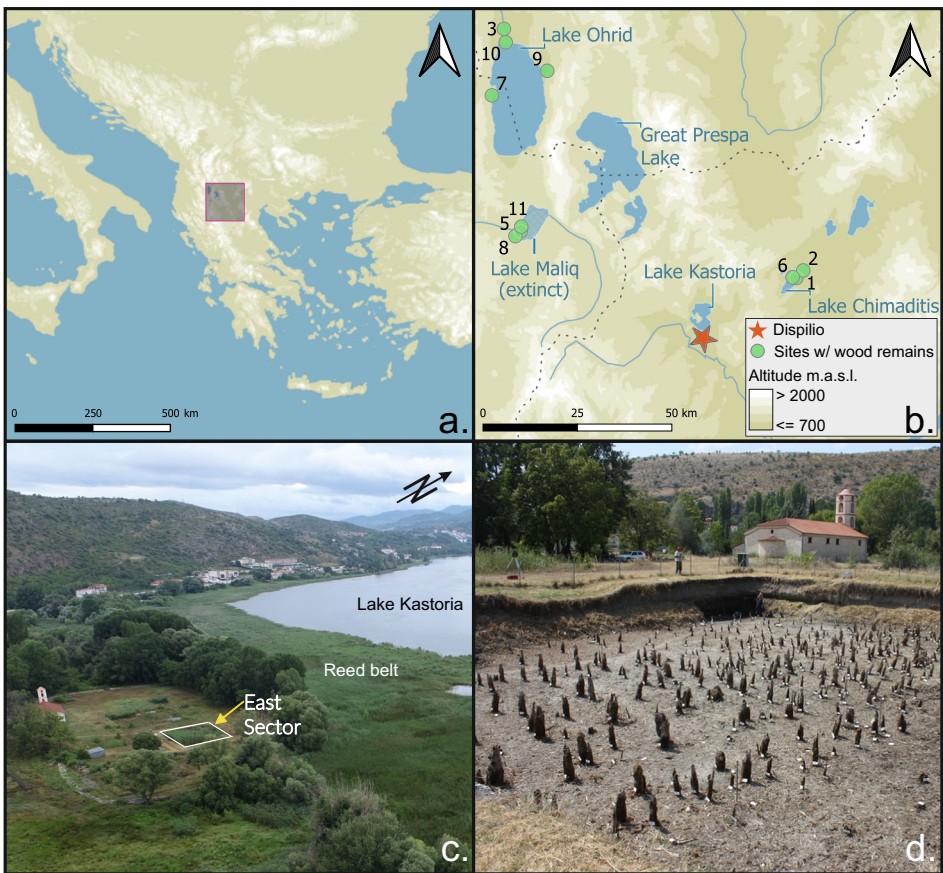

**Fig. 1 | Location of the archaeological site of Dispilio and detailed view of the trench analysed. a** map of S-E Europe highlighting the location of the enlarged area in **b**. **b** Location of Dispilio, and of other Neolithic sites within ~100 km with reported good wood preservation and similar chronological placement, thus holding significant potential for dendrochronological cross-dating with Dispilio. Sites depicted: 1-Anarghiri III; 2-Anarghiri IXb; 3-Crkveni Livadi; star-Dispilio; 5-Dunavec; 6-Limnochori II; 7-Lin 3; 8-Maliq; 9-Ohridati/Penelopa; 10-Ustie na Drim, 11-Sovjan; maps produced in QGIS 3.16, EPSG 32634, with SRTM[112] elevation data; Lake Maliq according to Fouache et al. [113]. **c** drone photograph of the site of Dispilio and its surroundings, the dendrochronologically analysed East Sector marked in the foreground. **d** close-up of the East Sector before sampling of wooden elements in 2019, vertical elements are seen projecting from the ground, each marked with a white label. (**a**, **b** A. Maczkowski; **c** M. Hostettler; **d** Dispilio Excavation Archive).

This limitation can now be overcome by a new hybrid form of dendrochronological and single-year radiocarbon analyses. Annual measurements of $^{14}$C in dendrochronologically dated Holocene tree-rings have revealed the occurrence of rapid short-term spikes in atmospheric $^{14}$C concentration in the past[19,20]. These $^{14}$C spikes, also called Miyake or solar energetic particle (SEP) events, are uniquely suitable for absolute dating of any wooden objects with detectable annual rings[9,21]. The discovery of these short-term events has also led to a proliferation of annual $^{14}$C measurements on single tree-rings, now spanning several millennia[22–24]. The mechanisms behind these $^{14}$C events are still debated[25,26]. However, a broadly accepted explanation is that they result from coronal mass ejections from the Sun[25,27–29]. These lead to a surge of SEPs colliding with the Earth's atmosphere, which increases the production of cosmogenic radionuclides[22,29]. To date, only five events[19,20,22,30] involving an atmospheric $^{14}$C increase of ≥1% within 2 years[22] have been confirmed. Of these, the two most recently discovered events are in the first half of the Holocene, in 7176 BC and 5259 BC[22], offering now the possibility for absolute annual dating of wood from the 8th and 6th millennia BC using annual $^{14}$C measurements.

In temperate climates, wood and other organic materials can be preserved only in very stable conditions, as in low-oxygen water-logged sediments at wetland archaeological sites[31–33]. Excavated wetland sites are very numerous and often excavated in Central Europe, and several wetland sites have also been found and excavated in south-eastern Europe, notably in the south-western Balkans[34–40]. Dendrochronological work on these sites has led to the construction of several tree-ring width chronologies, which were approximately fixed in time by $^{14}$C wiggle matching[41,42]. Dispilio on the shores of Lake Kastoria in north-western Greece is an archaeologically significant prehistoric wetland site in the region, considering that it is the only "pile-dwelling" settlement in the Balkans to be systematically excavated over multiple years and over a large area. Numerous lines of evidence have yielded detailed results on the site's geoarchaeology[39], palynology[43,44] anthracology[45,46], woodworking technology[47], and material culture[48,49] (Fig. 2). The approximate calendar-age chronology of the site has been established through radiocarbon dates, mostly performed on charcoal samples[39,50]. Charcoal is susceptible to the so-called "inbuilt age", which may produce overly old radiocarbon dates. The calibrated date-ranges point to settlement phases between the later Middle Neolithic (~5600 cal BC[51]) and the Bronze Age (~2100 cal BC[50]). The excavations at Dispilio have also yielded a great number of wooden remains, with over 1200 mapped construction elements in the site's East Sector to date (Fig. 1c). Yet despite the extensive remains of wooden construction elements, no systematic sampling and no dendrochronological studies have yet been conducted at the site. The value of developing a precise and accurate calendar-dated

**Fig. 2 | Archaeological finds from Neolithic Dispilio. a** almost completely preserved ornate anthropomorphic vessel from Late Neolithic. Many similar ones have been recovered from the site, scale in cm. **b** bone spear or harpoon tip with preserved hafting adhesives, scale in cm. **c** an assemblage of Late Neolithic personal adornments (**a**–**c**, Dispilio Excavation Archive).

chronological sequence using these wooden remains is further enhanced by the fact that the site of Dispilio has yielded more than 1700 complete and fully reconstructed ceramic vessels: one of the largest complete Neolithic ceramic assemblages in Europe. Tree-ring dating at Dispilio can therefore be used in conjunction with the ceramics typology network to underpin and improve the relative chronology of the entire region.

In 2019 a large-scale fieldwork campaign took place at Dispilio's Eastern Sector (Fig. 1d). Over 900 piles were mapped, of which 787 were sampled for the first dendrochronological analysis. The results provided a 'floating' oak tree-ring chronology spanning 120 years and an overlapping juniper tree-ring chronology spanning 303 years. However, this dendrochronological record could not be dated absolutely because despite the existence of several millennia-long tree-ring chronologies in the Eastern Mediterranean[17,52,53], none extend back to 7500 years. Here, we overcome this limitation by using the combination of dendrochronological and single-year radiocarbon analysis (Fig. 3), placing the last ring of the Dispilio juniper chronology at 5140 BC. To our knowledge, no calendar-year absolute dating has been achieved yet for any other Neolithic site in the wider Mediterranean region.

## Results
### Dendrochronology
Wood samples from Dispilio were first sorted by genus, then measured and cross-dated into tree-ring width (TRW) chronologies. Then, individual tree-rings from the cross-dated wood samples were sampled for [14]C measurements to identify the 5259 BC [14]C spike in the juniper tree-ring chronology, which provides the absolute calendar-year placement of the tree-ring chronology. Of the total wood samples from the 2019 fieldwork ($n = 787$), 23% were cross-dated into two master tree-ring width (TRW) chronologies. Wood anatomical species determination revealed that the majority of the wooden piles came from juniper (*Juniperus* spp., 62%) and oak (*Quercus* spp., 21%) wood. The third most abundant genus was pine (*Pinus* spp., 17%), which was not suitable for dendrochronological cross-dating given the low number of annual rings on most pine samples. Due to the wood-anatomical intra-genus similarity of junipers[54,55], and of deciduous oaks from the subgenus *Quercus*[56], a definitive species-level identification was not possible. Based on the distribution of modern tree species in the region[45,57,58], Dispilio oak wood samples most likely come from *Quercus frainetto*, *Q. petraea*, and/or *Q. pubescens* wood, and the junipers are most likely *Juniperus excelsa*, *J. foetedissima*, and/or *J. deltoides* (cf. *J. oxycedrus*).

The oak TRW chronology extends over 120 years and is composed of 58 wood samples (Fig. 4). It consists of tree-ring sequences with an average segment length of 66 years. Some sapwood was present on most of the oak samples ($n = 45$); however, the final growth ring or "waney-edge", which is important for archaeological interpretation, was conserved on only four pieces, as a result of either the lower durability of oak sapwood or its intentional removal. The mean inter-series correlation[59] of the oak tree-ring sequences is 0.51.

A 303-years-long juniper TRW-chronology was also constructed with 118 tree-ring sequences and an average segment length of 86 years (Fig. 4). The mean inter-series correlation[59] of the juniper chronology is 0.62. Juniper wood's chemical[60] and physical[61] properties give it high resistance to degradation. These qualities made juniper wood the material of choice for construction purposes in many ancient societies in the Eastern Mediterranean[62–64]. The preservation of juniper wood in Dispilio is also exceptional and the waney edge on junipers is quite common, enabling an annually resolved reconstruction of the building phases and occupation duration on the site (Fig. 4b).

All samples with a preserved waney edge had a final growth ring terminating with latewood, thus implying a felling date during the dormant period of the trees between late summer/autumn and early spring. The juniper and oak TRW chronologies have robust dendrochronological dating against each other (*t*-value = 4.9[65] and =5.1[66]; GLK = 63%[67]) over a period of 108 years where sample replication is >4. This dendrochronological cross-dating between the two chronologies, considering that the oak chronology does not span the Miyake event, is independently supported by a conventional [14]C wiggle-matching model of 11 [14]C measurements performed on tree-rings comprising the oak TRW chronology[51] (Supplementary Note 1).

### Tree-ring [14]C cosmogenic signature and calendar-year dating
The preliminary modelled end-date range for the juniper tree-ring width chronology was established through a conventional radiocarbon wiggle-matching model[68,69], with the IntCal20[70] radiocarbon calibration curve (see Supplementary Note 1). On the basis of this coarse resolution wiggle-match dating we identified the wood samples and their corresponding tree-rings that should approximately span the period of the 5259 BC Miyake event. Four individual juniper wood samples were selected for annual [14]C measurements, and the tree-rings approximately spanning the predicted position of the 5259 BC Miyake event were dissected at annual resolution (Fig. 3a and Supplementary Figs. 1-5). Here, we present the 115 single tree-ring [14]C measurements (Supplementary Data 1) performed to locate the 5259 BC Miyake event in all four wood samples selected from the master Dispilio juniper tree-ring chronology (Fig. 3a). An average year-to-year increase (*sensu* Miayke et al. [19]) of -15.8 ‰ in Δ[14]C was detected in all wood samples (Fig. 3a and Supplementary Data 1) in the exact same dendrochronologically cross-dated tree-rings corresponding to the relative year 184 of the Dispilio juniper chronology. This increase varies from the lowest of -11.1 ‰ Δ[14]C in wood sample DISP-10070, to -13.1 ‰ in DISP-10206, to -14.8‰ in DISP-10063, and to -18.6 ‰ in DISP-10611 (Fig. 3a and Supplementary Data 1).

To compare the [14]C results from Dispilio with the published reference data for the 5259 BC event, a mean-value annually resolved reference curve (RC) was established from the dataset in Brehm et al.

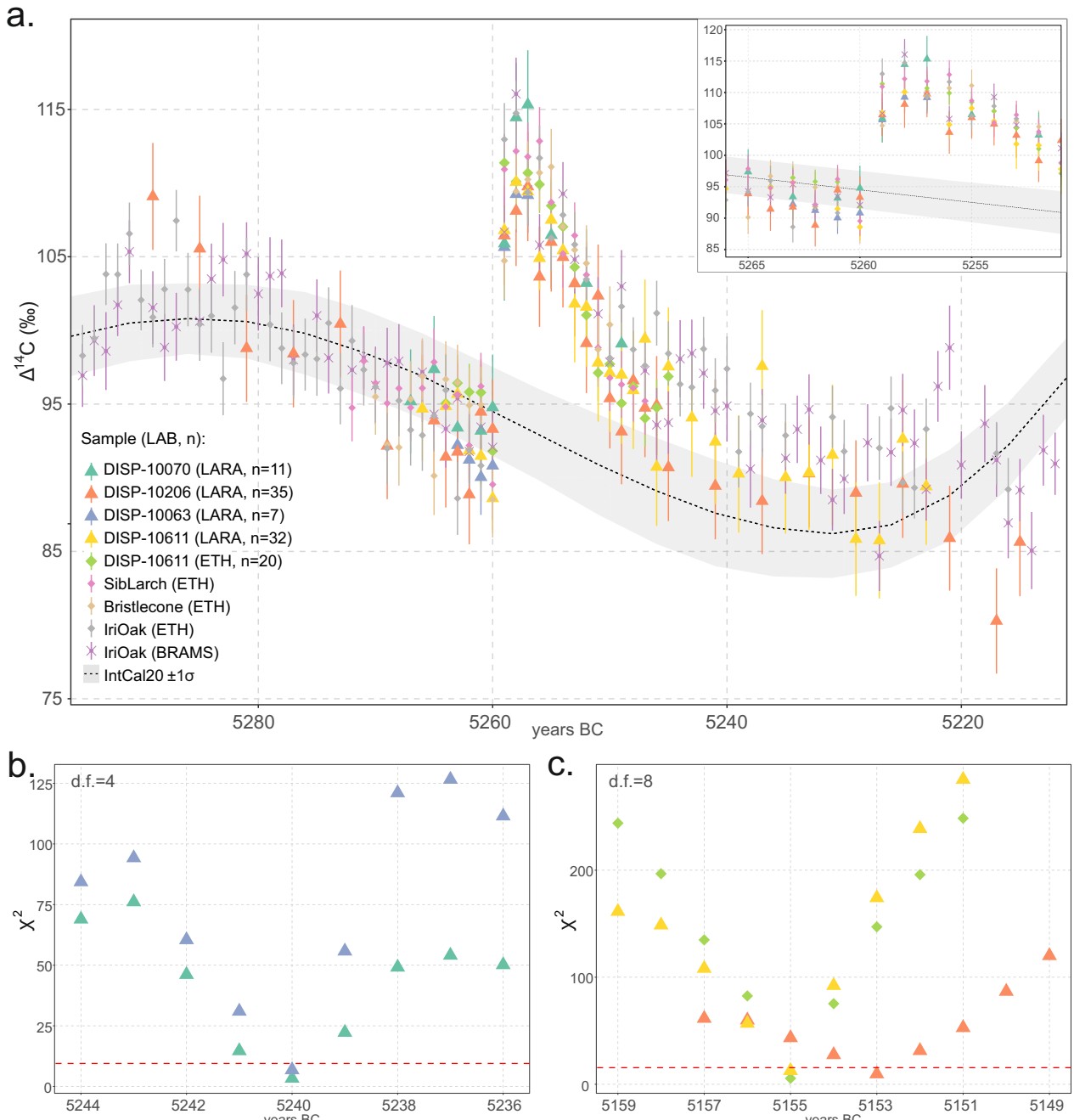

**Fig. 3 | Scatter plot of $\Delta^{14}C$ data from Dispilio against reference from Brehm et al.[22] and IntCal20[70], and best last ring-fit for the dated wood samples ($\chi^2$).**
**a** Measured $^{14}C$ concentrations represented as $\Delta^{14}C$, vertical bars represent $1\sigma$ errors (Supplementary Data 1); legend labels marked with "DISP-" and larger symbols refer to wood samples from Dispilio and corresponding $^{14}C$ measurements obtained in this study; other labels represent data from Brehm et al.[22], measurements from Siberian larch, Bristlecone pine, and Irish oak measured at two different labs, symbol shapes according to Lab; Bristlecone pine $^{14}C$ data are shifted forward by 1 year from the original BR22 publication following a correction to the dating of

the bristlecone master tree-ring chronology (Supplementary Note 2); shaded band represents IntCal20[70]. Inset in right corner same as in main panel **a**. but at higher resolution, spanning 14 years centred around the $^{14}C$ spike. Lower two panels: $\chi2$ tests of Dispilio measurements against the average from BR22[22], **b**. $\chi^2$ test results for wood samples DISP-10,070 and DISP-10063 ($\chi^2$ crit. value = 9.49), and **c** for wood samples DISP-10206 and DISP-10611 ($\chi^2$ crit. value = 15.51), legend as in panel above. Figure panels produced in R[101], code and source data available in Supplementary Data 3.

(2022: henceforth referred to as 'BR22'[22]). A common approach to verifying the position of Miyake events is wiggle-matching using a goodness-of-fit $\chi^2$ test[9,10,71] against a reference, so that the $\chi^2$ value becomes minimal for the correct placement of the sample's waney-edge[68]. The lowest $\chi^2$ values are reached when the end-dates of the wood samples are placed at 5240 BC for DISP-10070 and DISP-10063 (Fig. 3b), 5153 BC for DISP-10206, and 5155 BC for DISP-10611 (Fig. 3c),

corresponding to their cross-dated position along the tree-ring chronology. The 5259 BC event signal is clearly identified in all wood samples (Fig. 3a).

To test how close conventional radiocarbon wiggle-matching would be relative to the absolute calendar dating supplied by the Miyake event, the annual data from all four wood samples were wiggle-matched against the IntCal20 calibration curve[70] using the

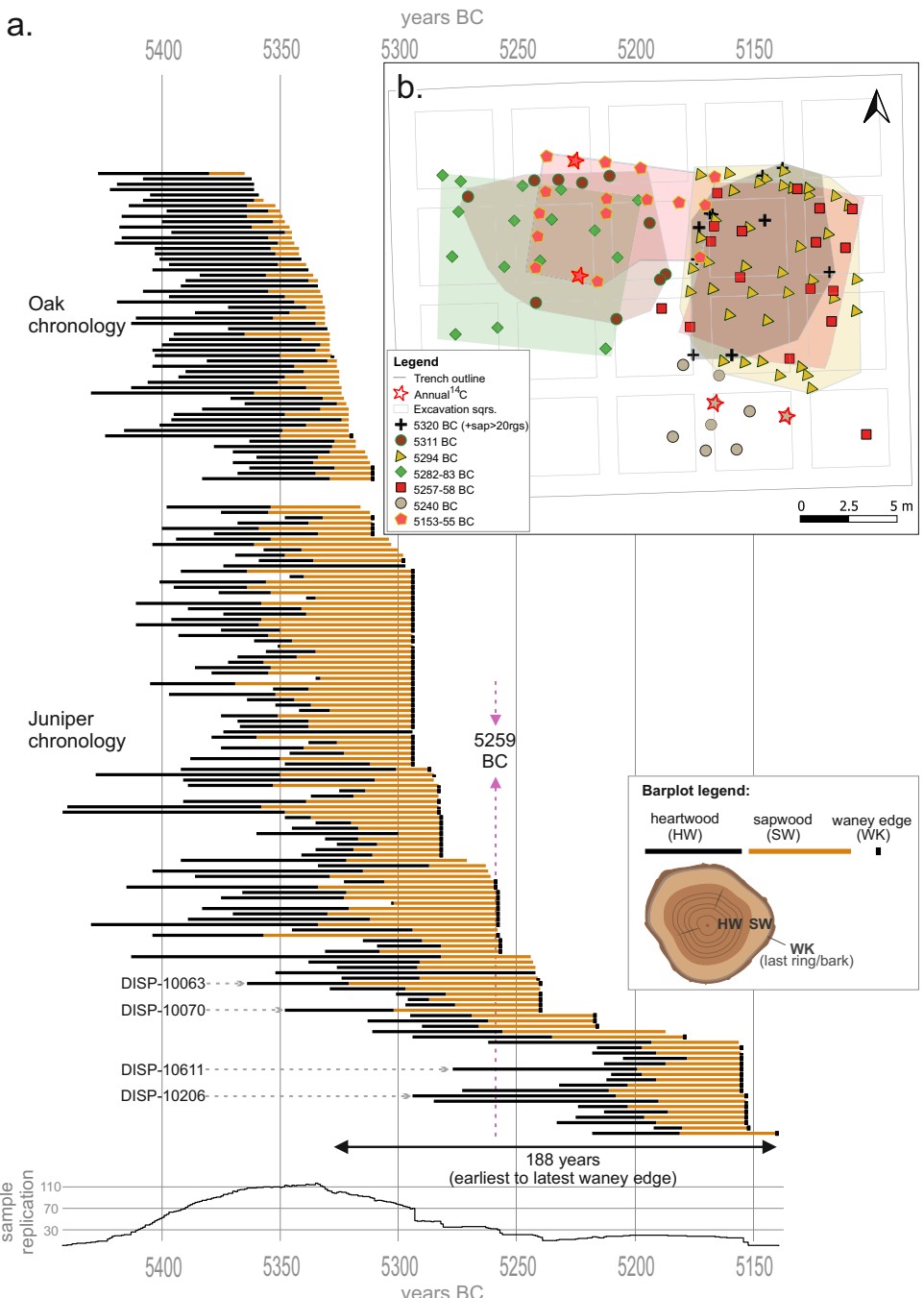

**Fig. 4 | Bar chart of tree-ring chronologies, felling dates, and archaeological site plan development of the East Sector, Dispilio, Greece. a** bar plot of Dispilio oak and juniper chronologies, calendar-year dating as obtained by identifying the radiocarbon spike of 5259 BC[22] in the juniper chronology; each horizontal bar represents an individual wood sample in its dendrochronologically cross-dated position; bar length corresponds to its span in years measured by number of tree-rings; bar plot legend with the drawing of a wood sample cross-section schematically describes the meaning of each horizontal bar; "DISP-…" labels and dashed lines with arrows point to the location in the tree-ring chronology of the wood samples sampled for annual [14]C (Supplementary Figs. 1-5). **b** schematic plan of the East Sector (see also Fig. 1c, d); each symbol represents one vertical wooden element, different shapes and colours correspond to the same felling phase spread over 1–2 years; additionally, the colour-shaded polygons outline the groups of same symbols (same felling-phase wooden piles), however they do not represent definite structure plans; red stars represent the location on the site plan of the wood samples in which the radiocarbon spike of 5259 BC was identified.

OxCal 4.4 [14]C calibration software[68,69]. When IntCal20 is used, in none of the cases does the 95% probability end-date range include the actual felling date (Fig. 5 and Supplementary Data 3). Longer series of [14]C dates that span some years before and after the event (Figs. 3a and 5), such as from wood samples DISP-10611 and DISP-10206, yield end-dates that are only ~15–20 cal years older, and shorter series, such as from wood samples DISP-10070 and DISP-10063, result in end-dates over ~40 cal years younger than the actual felling dates (Fig. 5). It has been noted previously[72] that IntCal20 is poorly replicated during the 53rd and 52nd centuries BC. Notably, the 53rd century BC is represented by only 16 measurements, of which 14 are decadal and bidecadal blocks of 10–20 tree-rings, with only two 4-year and 5-year blocks[70,73] (Supplementary Fig. 8). The variability in the calibrated end-date ranges suggests that IntCal20 might produce

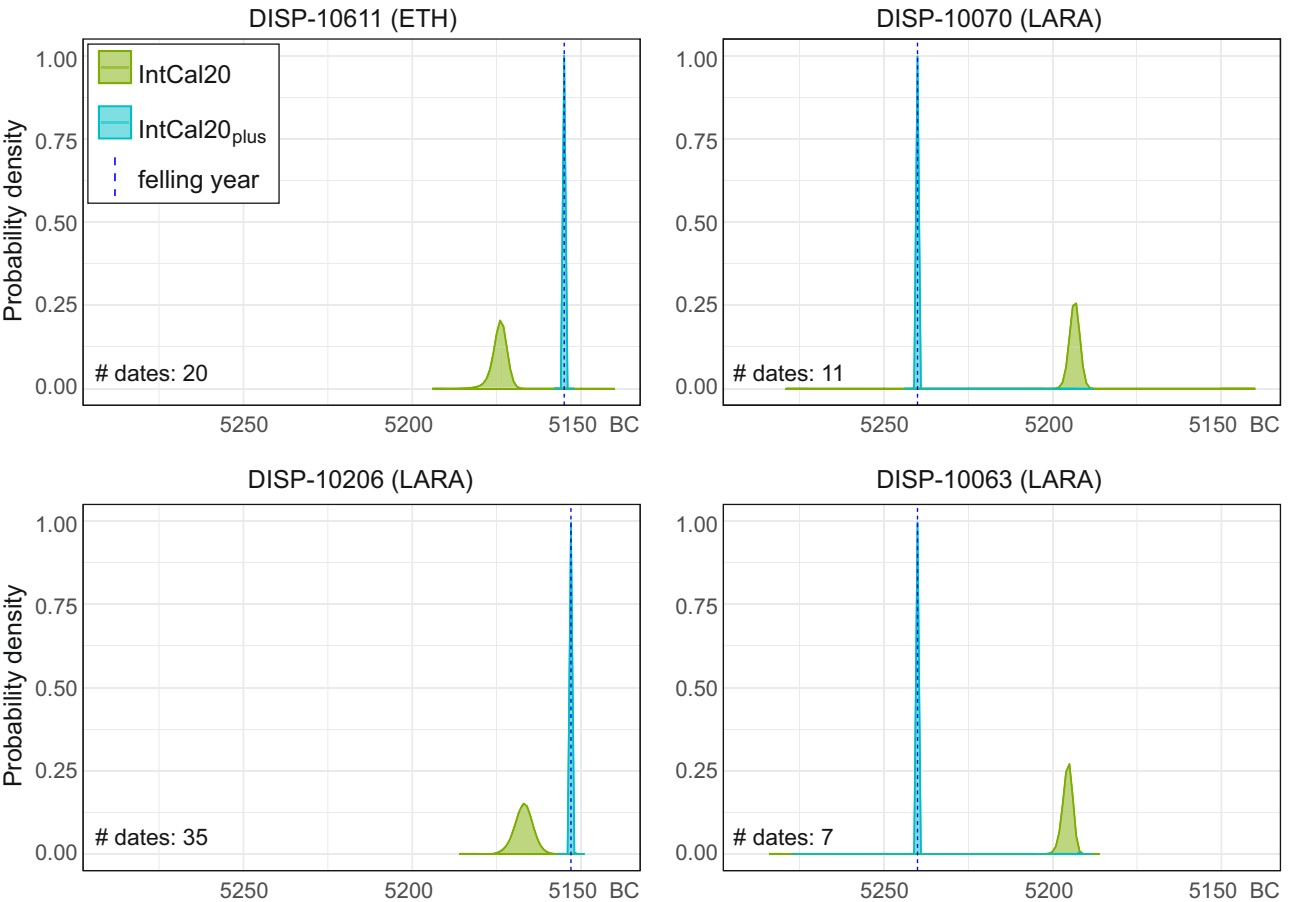

**Fig. 5 | Probability distributions of end-date ranges produced by wiggle-matching different sets of annual 14C data from Dispilio modelled in OxCal v4.4, against IntCal20[70], and IntCal20plus.** In IntCal20plus the non-annual IntCal20 data for a 82-year period around the 5259 BC Miyake event is replaced by the annual average of Brehm et al.'s.[22] annual data. Dashed blue lines represent actual felling dates determined through dendrochronology and Miyake event-matching. Acronyms in brackets next to sample name refer to the AMS lab that furnished the measurements. Data for figure obtained from OxCal 4.4[68,69]. Figure produced in R[101]. R code, OxCal code and source data in Supplementary Data 3.

misleading results when wiggle-matching annual data coming from the period in question. The annual 14C dates were also wiggle-matched against a modified IntCal20, IntCal20plus, in which the default IntCal20 multiple-year blocks of before-present (BP) data for the 82-year period around the event were substituted with the average of the annual BR22 dataset. Calibrating against this dataset predictably yields the accurate and more precise end-date ranges at 95% probability for all wood samples (Fig. 5).

The growing season of trees is influenced by many factors and can vary between and within species as a function of cambial age, temperature, water availability, slope, aspect, soil type, and other factors. Personal observations of growth termination in modern oaks and junipers in the region have shown that latewood width at breast height can be completed in both genera by the beginning of September (Supplementary Figs. 6 and 7). While cell-wall thickening in temperate conifers continues for several weeks after the cessation of cell-wall enlargement[74], the amount of cellulose carbon that would be deposited during this last stage of latewood formation constitutes a small percentage of the whole tree-ring[75]. Considering the robustness of the 14C signal in the Dispilio junipers tree-rings (Fig. 3) it is unlikely that it only represents the 14C incorporated at the end of the cell-wall thickening stage. Consequently, it can be inferred that the 14C signal of the 5259 BC event in the indeciduate junipers has been incorporated during the temperate growing season between spring and late summer/early autumn 5259 BC.

According to the dendrochronologically cross-dated position of all wood samples, the ring in which the Miyake event is detected corresponds to relative year number 184 of the 303-year-long juniper TRW chronology. This allows us to set the absolute end-date of the whole Dispilio juniper tree-ring chronology at 5140 BC. Furthermore, the identification of the event in wood samples DISP-10070 and DISP-10063 confirms the correct placement of the better-replicated earlier half of the chronology (Fig. 4a). Given the dendrochronological cross-dating between the juniper and oak chronologies, also the latter is absolutely dated, placing its last ring at 5311 BC (Fig. 4a.).

### Site plan and felling phases

The timespan of 188 years between the earliest and latest felling dates (Fig. 4a) indicates the minimum duration of construction activities during Dispilio's 54th–52nd centuries BC settlement phases. This timespan between 5328 and 5140 BC includes intermittent periods of wood felling and construction (Fig. 4a, b). However, the timespan between the earliest and latest felling dates in Dispilio need not alone reflect a continuous, uninterrupted occupation of the same location.

Wooden piles that have been dendrodated against the Dispilio juniper master chronology with felling dates within 1–2 years of one another, were plotted on the site plan using a GIS software (Fig. 4b). This revealed outlines that probably represent different structures (Fig. 4b). Identification of building outlines was possible only for groups that are composed of a substantial number of dendrodated

samples. The structures seem to be oriented along the modern lakeshore; however, the shape of the Neolithic lakeshore is uncertain. Of note is the concentration of building activities in the eastern part of the Easte Sector (Fig. 4b). In this part, a group of wooden piles with a felling date of 5294 BC outline an area, superimposed by a group of piles with felling dates in 5258–5257 BC (Fig. 4a, b). The juniper group felled in 5294 BC is preceded by a potential felling phase ending in 5320 BC consisting of oak wood samples. However due to the poor preservation of oak sapwood only two of this group have preserved waney edge. These two securely dated piles are complemented by several other oak piles with final measured rings falling between 5328 BC and 5320 BC and no waney edge, but with at least 20 sapwood rings indicating the proximity of the waney edge. The mapping of the dendrochronological results also implies that building practices in some cases included either the short-term storage of timber for 1–2 years or a construction period spread over several years.

## Discussion

No absolute timeframe has been generally agreed for the archaeochronological periodisation of the Neolithic in the region (e.g. refs. 8,76). Therefore, depending on the sources consulted, the occupation phases of Dispilio discussed here would fall in the later Middle Neolithic and/or Late Neolithic. In this context, the absolute calendar-year dating of Dispilio is a step forward in establishing a more readily navigable periodisation of the Middle and Late Neolithic in the region[8,76]. The precision of the absolute dating and duration of the 54th-52nd century BC occupation and construction phases in Dispilio is unique not only in the Neolithic of the Balkans, but also in the wider Eastern Mediterranean. The site also provides sufficiently replicated dendrochronological information to serve as an independent control for settlement durations which are mostly estimated from modelled [14]C dates[77,78].

The felling dates in the excavated sector indicate activity over a period of at least 188 years, and indications from oak sapwood reconstruction estimates may extend this backwards in time by a further ~30 years. Of particular interest is the succession of two construction phases in the western half of the analysed trench and three construction phases in its eastern half (Fig. 4a, b). Although the potential function of these structural outlines (Fig. 4b) is not clear at present, the precise timespans between the construction episodes of 29 years in the western half (5311 and 5282 BC), and 35–37 years in the eastern half (5320, 5294 and 5257 BC) are consistent with the few available estimates of house lifespans in Neolithic S-E Europe[77,78]. However, determining whether these contemporary structure outlines with similar felling dates correspond to one building or more will require further detailed multidisciplinary work. Intermittent periods without felling dates may simply be a result of worse preservation of some elements or the limited size of the excavated area, but they may also reflect a hiatus in occupation, or indicate that settlement was of a nonperennial character. Detection of annual or decadal-scale hiatuses is extremely difficult in archaeological stratigraphy, with settlement phase duration usually estimated from [14]C sequence models based on organic samples from consecutive stratigraphical units. This approach can lead to interpretations of centuries-long settlement continuities[4,79]. Such interpretations may underestimate settlement discontinuities of durations shorter than the precision of [14]C measurements and calibration. This underlines the importance of the annually resolved data from Dispilio.

The last centuries of the 6th millennium BC mark an important change within the Neolithic period in the southern Balkans. It is a period of a steep increase in the number and size of settlements, associated with a demographic boom[8,80–82]. Anthropogenic influence on the local environment becomes pronounced during this period[83,84], as also documented at Dispilio[43,44]. Diversity increased in all aspects of human behaviour, from pottery production techniques and styles[85],

architecture[81], settlement organisation[81,86,87], to the first signs of metallurgy[88]. Evidence from this period also indicates a shifting social focus from the collective to the domestic[89,90], when individual households seem to play a more prominent role in community life. In this setting, high-resolution chronological data can improve our understanding of societal changes, human land use, and intensifying influence on the local and regional environment. For instance, the preference for settling in the proximity of wetlands has been documented in the Early Neolithic[3,91], and the practice continued in the Middle and Late Neolithic[36,91]. Wetland and shoreline locations would have represented ideal catchment areas for the Neolithic subsistence, providing various soil types that could be used for cultivating crops with different requirements, serve as pasture lands, or supply aquatic resources as a dietary complement[91]. A number of wetland sites with similar chronology to Dispilio's phases discussed here (2nd half of the 6th millennium BC) have been documented or excavated in existing or former lakes in the region (Fig. 1b), some of them yielding large amounts of well-preserved wooden construction elements[36–38,92–94]. Although the dating of these sites has much lower chronological resolution than at Dispilio, some of them would have been in use for centuries before and/or after the 54th–52nd century BC phases at Dispilio. It is highly likely that it will be possible to cross-date the tree-ring widths of the wood remains from these neighbouring sites with the now absolutely dated tree-ring chronologies from Dispilio, and thus extend the absolutely dated dendrochronological network for the region well beyond the 6th millennium BC.

Beyond the chronological significance, absolutely dated tree-ring records are one of the most frequently used proxies for high-resolution climate reconstructions and offer unique insights into the relationship between societies and climate. Precipitation is a limiting factor for most low and mid-altitudes trees in the Eastern Mediterranean. In fact, modern juniper[53] and oak[17,95] tree-ring sequences have been shown to be good predictors of precipitation in the Eastern Mediterranean. Precipitation was a crucial factor in early agriculture, which mainly consisted of rainfed[96] and floodwater[97] farming. Preliminary observations of the Dispilio TRW chronologies indicate a period of suppressed growth in both the juniper and oak tree-ring sequences for a period of around 20 years between 5360 and 5340 BC. Such a period of suppressed growth can be associated with a decrease in precipitation, which may have influenced the water table of small water bodies such as Lake Kastoria. A short-term Mid/Late Neolithic eutrophication of the lake previously inferred from the increased occurrence of green algae[39] could potentially be correlated with this tree-ring width suppression. Although the Neolithic tree-ring sequences from Dispilio are relatively short if compared to modern tree-ring proxies used in climate reconstructions, they may still provide valuable, absolutely dated, annually resolved information on environmental conditions during the Neolithic in Kastoria Basin and the surrounding region.

Finally, the results from this study underline the value that single-year measurements of radiocarbon in tree-rings can have for radiocarbon calibration and dendrochronological dating. Significant advances in AMS technology[98], have enabled the creation of continuous time-series of annual radiocarbon measurements that are constantly improving the resolution of the radiocarbon calibration curve, thereby increasing the accuracy of the radiocarbon calibration process. Moreover, the value of SEP events in anchoring regional timelines through hybrid tree-ring and radiocarbon studies is once again demonstrated. The [14]C-anchored Dispilio tree-ring chronologies now provide a calendar-dated reference for dendrochronological dating of other sites from the time period. This provides the opportunity to extend calendar-dated chronologies across the region further back into prehistory. Such high-resolution dating, especially when it can be coupled with stratigraphic information or used to derive climatic indicators, will elucidate a more

nuanced understanding of deterministic interpretations of the environmental influence on societies in the past, for instance for the 6.2 ka BC cooling event. This study demonstrates how the recent discovery of the SEP events in this time period creates new possibilities in prehistoric archaeology and offers the construction of historical-timescale narratives for societies and their environments from the very distant past.

## Methods
### Wood samples
The wood material analysed in this study was sampled in August and September 2019 from wooden piles remains at the archaeological site of Dispilio, near Kastoria, Greece (40.485444 N, 21.289694 E; h = 627 masl). The site is one of the best-known prehistoric sites in the country and has been investigated, almost continuously, since 1992. Excavations and sampling that took place on the site were performed in full compliance with the regulations of the Greek Ministry of Culture concerning archaeological material. The permit number for the wood samples obtained from the Ministry is: ΥΠΠΟΑ/ΓΔΑΠΚ/ΔΣΑΝΜ/ΤΕΕ/ Φ77/379195/266411/4122/252, issued on 24.07.2020. During the 2019 fieldwork campaign, whole cross-section discs (n = 787) were sampled from the wooden remains with handsaws and chainsaws. The wood samples' documentation, cleaning, preparation, and sealing in plastic bags with water, took place on-site. Dendrochronological measurement began on-site and continued at the University of Bern. Tree-ring width (TRW) measurements were performed according to standard dendrochronological procedures[99,100], with a measuring table under a binocular stereo microscope. TRWs were recorded with a precision of 0.01 mm. Two to four radii were measured per sample and averaged together to represent the sample. Descriptive dendrochronological statistics were performed in the dplR package in R[59,101,102]. The TRW measurements of DISP-10611, -10206, -10070, and -10063 are available in both Source Data and Supplementary Note 2. All wood specimens will be permanently deposited at the Dispilio Excavation Laboratory, Dispilio, Greece. Access is subject to obtaining permits from the Greek Ministry of Culture.

Wood taxonomy was determined from stem wood anatomy. Each measured wood sample was sectioned with a razor blade and cell arrangements in the transversal, radial, and tangential sections were identified and compared with references in wood-anatomical atlases[55,56,103]. Given the wood-anatomical similarity of various deciduous oak species from the subgenus *Quercus*[56], and considering the high dendrofloristic diversity of oaks in the region[57,58], it was not possible to distinguish them to species level. However, several deciduous oak species from the subgenus *Quercus* are likely to be represented, notably *Q. frainetto*, *Q. petraea*, and/or *Q. pubescens*. Oak trees from the subgenus *Cerris* are one of the more abundant groups of oaks in the region; however, no wood samples from Dispilio could be assigned to this group, which is anatomically characterised by larger and solitary latewood pores. Similarly, wood anatomical differentiation between different juniper species was not possible[54,55,103]. Considering today's distribution of tree-like junipers in the region, the species most likely to have been used in Dispilio are *Juniperus excelsa*, *J. foetedissima*, and/ or *J. deltoides* Adams (cf. *J. oxycedrus* L.). Although the majority of the pine samples exhibited denticulate walls on end-tracheids, a characteristic of the pine subgenus *Pinus* (cf. *Pinus nigra/sylvestris*-type), several pine wood samples could be identified as members of the subgenus *Strobus* (cf. *P. peuce*) by the presence of smooth-walled end-tracheids.

Modern local climatic conditions in the Kastoria Basin can be defined as continental to sub-Mediterranean, with temperate weather, continental winters, and warm and dry summers. The yearly average precipitation in the lower parts of ~700 mm increases with altitude. The wettest months are November and December, and the driest and hottest months are July and August. Yearly average temperature is

-12 °C. Main climate classes according to the Köppen system[104] are Cfa, Cfb, and Csa.

### Sample preparation and radiocarbon measurement
Individual tree-rings were dissected by hand under a binocular microscope with a one-sided razor blade (Supplementary Fig. 5). Whole rings were used for all $^{14}$C measurements (Supplementary Data 1). About 30–70 mg of material was sampled per ring, depending on its width. Earlywood comprises ca. 80–90% of a juniper tree-ring. Because most of the of the ring-width of junipers growing on mesic sites is completed by the end of September[105] (see also Supplementary Figs. 6 and 7), the tree-ring structural carbon concentration should reflect temperate spring-to-late summer carbon uptake.

Wiggle-matching of 11 $^{14}$C dates from the juniper chronology (modelled end-date 5233-5137 cal BC, 95% probability, Supplementary Note 1) provided the basis for an initial estimate of the segment on the tree-ring chronology where the event was to be located. A series of 70 individual tree-rings centred around an estimated event ring were dissected from the first wood sample that was analysed (DISP-10206, Supplementary Fig. 1). The $^{14}$C content of every 4$^{th}$ sampled ring was subsequently measured until the $^{14}$C spike was identified, after which the $^{14}$C in 20 consecutive rings around the event was measured (Supplementary Data 1). The event ring on all the other wood samples (DISP-10611, DISP-10070, and DISP-10063, Supplementary Figs. 2-4) was identified according to the samples' dendrodated position along the tree-ring chronology.

Cellulose from wood samples analysed at the Laboratory for the Analysis of Radiocarbon with AMS at the University of Bern (LARA)[106] was extracted following the BABAB method[107] including Sookdeo et al.'s. [98] modifications at 70 °C for all steps. Samples were submerged in a 1 M NaOH overnight and treated in 1 M HCl followed by 1 M NaOH in a shaker for 1 h each. Bleaching of the samples was performed on addition of 5 mL water, a few drops of 1 M HCl to reach pH 2–3 and 100 mg NaClO2 by shaking for at least 2 h or until the colour of the wood samples turned white. The material was dried by lyophilisation overnight. Samples were measured using the LARA MICADAS AMS system. In June – November 2022 the tree-rings from wood samples DISP-10070, -10206 and a first run of -10611 were analysed together with three oxalic acid II (SRM 4990 C, NIST) standards and three chemical blanks. In June 2023, DISP-10063 and a second run of DISP-10611 were dated together with five oxalic acid II standards and four chemical blanks that were used for blank subtraction, standard normalisation, and correction for isotope fractionations. Two IAEA-C5, two IAEA-C7, two 1515 CE reference samples and two cellulose blanks were also used as secondary standards and blanks, respectively. All initial results (Supplementary Data 1) were biased by an inappropriate batch of the oxalic acid II standard that was used for the measurements. A total of ten samples from our first run of DISP-10611 were repeated within our second run which gained an average negligible difference between both LARA runs of 0.6‰ so that our DISP-10611 dataset that is shown Fig. 3a represents the average of both LARA runs (Supplementary Data 2). By intercomparison with three other oxalic acid II batches (one of which was provided by ETHZ), an offset corresponding to a $^{14}$C age of +30.9 ± 3.2 years was determined for the inappropriate batch, whereas the results of the oxalic acid II of the other three batches were identical within uncertainties. The initial results were corrected for this offset yielding a shift of ~+4.2‰ for the samples. The corrected results are presented in Fig. 3a.

For the analyses performed at the Laboratory of Ion Beam Physics at ETH Zürich (ETH), the tree-ring samples were prepared in 15 ml glass test tubes together with four wood blanks (2 BC and 2 KB) and 2 1515 CE reference samples each weighing 30–60 mg[98,108]. In a slightly modified procedure following[107], samples were first soaked in 5 ml 1 M NaOH overnight at 70 °C in an oven. Then the samples were treated with 1 M HCl and 1 M NaOH for 1 h each at 70 °C in a heat block, before they

were bleached at a pH of 2–3 with 0.35 M NaClO2 at 70 °C for 2 h. The remaining white holo-cellulose was then freeze-dried overnight. -2.5 mg dried holo-cellulose was wrapped in cleaned Al capsules and converted to graphite using the AGE-3 automated graphitisation line. A measurement set was assembled comprising the tree-ring samples, three oxalic acid one (OX1) and four oxalic acid two (OX2) standards, two cellulose blanks, two chemical blanks, and two 1515 CE reference samples and measured in the MICADAS accelerator mass spectrometer.

### Radiocarbon matching and modelling

The $^{14}$C measurements presented in this study were matched to the constructed reference curve[22] (Supplementary Data 3) using a common $\chi^2$ test approach (Eq. (1)) so that the $\chi^2$ value becomes minimal for the correct placement of the sample's waney-edge[9,10,68]:

$$\chi^2(x) = \sum_{i=1}^{n} \frac{(R_i - C_{(x-r_i)})^2}{\delta R_i^2 + \delta C_{(x-r_i)}^2} \qquad (1)$$

here $R_i \pm \delta R_i$ represent the new $^{14}$C measurements, $C_{(x-r_i)} \pm \delta C_{(x-r_i)}$ represent the reference curve $^{14}$C concentrations in the year $(x - r_i)$; $x$ is the assumed calendar year; and $r_i$ stands for the tree ring number starting with 0, representing the last growth ring of the tree (waney edge).

The Bayesian wiggle-matching was performed in the OxCal 4.4 software with the D_Sequence command against the atmospheric data from IntCal20[69,70]. For the CQL code see Supplementary Note 1, and the end of the.Rmd file in Supplementary Data 3.

The year-to-year increase in $\Delta^{14}$C presented in the Results section was calculated as a difference between the values in 5260 BC and 5259 BC (*sensu* Miyake et al.[19]). For a detailed discussion on the magnitude and $^{14}$C production during the 5259 BC Miyake event see[22] and[109].

Calendar years are expressed according to the Gregorian calendar (AD/BC), without year 0.

### Data uncertainty

The genus *Juniperus* is known to produce intra-annual density fluctuation ('false rings'), and to have 'missing rings'[110] in parts of the stem. Missing rings are very often a product of the stem growth habit of junipers, termed 'lobate growth', which involves higher cambial activity and faster growth in certain areas of the stem, resulting in an undulating cross-section of the stem in older tress, where the less active areas may not produce rings in certain years. However, missing rings or measuring false rings can be accounted for when sufficient numbers of wood samples with complete stem cross-sections are available, as in Dispilio. The correct location of the "event ring" on all wood samples determined from their cross-dated position further supports an accurate ring count. Moreover, the dendrochronological cross-dating of the first half of the juniper chronology against the oak chronology provides an additional control for the correct ring count, because oak trees hardly ever have missing rings[111].

### Reporting summary

Further information on research design is available in the Nature Portfolio Reporting Summary linked to this article.

## Data availability

Supplementary Data 1-2, Supplementary Information including Supplementary Notes 1-2 and Supplementary Figs., as well as Supplementary Data 3 for this paper are available both at the journal's website, and also at the following repository: https://doi.org/10.5281/zenodo.8407222 [https://zenodo.org/records/10981405]. The use of DOI resolver is recommended for being directed to the latest version of the Supplementary Information. Individual tree-ring width measurements of wood samples not discussed in the text are available on request, as they will form part of an upcoming dedicated publication. Source data are provided with this paper.

## Code availability

OxCal code is available in the Supplementary Information file in the section Supplementary Note 1, and at the bottom of the.rmd file in the Supplementary Data 3.

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

## Acknowledgements

The 2019 fieldwork and the subsequent dendrochronological and radiocarbon analyses were conducted in the framework of the 'Exploring the dynamics and causes of prehistoric land use change in the cradle of European farming' (EXPLO) ERC project. This project is financially supported by the European Union's Horizon 2020 research and innovation programme, under the grant agreement No 810586 (project EXPLO, exploproject.org). We would like to thank all the archaeology students involved in the fieldwork and sample curation from the Universities of Thessaloniki and Bern, and the staff of the Ephorate of Antiquities of Kastoria. Thanks are due to Ariane Ballmer for arranging the logistics of the wood sample transport.

## Author contributions

A.M., together with C.P. and A.H., conceived and designed the study. K.K. and T.G. led the fieldwork, and A.M. and J.F. participated in part of it. J.F. and A.M., together with M.B., performed the dendrochronological and wood-anatomical analyses. A.M. sampled individual tree-rings. S.S. and L.W. performed and provided the 14C measurements. A.M., together with C.P., drafted the manuscript, and all authors edited and contributed to the manuscript. A.H. and K.K. obtained funding.

## Competing interests

The authors declare that they have no known competing financial interests or personal relationships that could have appeared to influence the work reported in this paper.
