## [Peer Review File · Nature Communications]

Absolute dating of the European Neolithic using the 5259 BC rapid 14C excursionReviewers' Comments:

Reviewer #1:

Remarks to the Author:

This study at Dispilio revealed robust oak and juniper tree-ring chronologies, providing a 120-year oak TRW chronology and a 303-year juniper TRW chronology. The detection of the 5259 BC Miyake event in the juniper chronology allowed for the absolute dating of both chronologies, placing the end-dates at 5140 BC and 5311 BC, respectively. Additionally, the findings suggested a 188-year minimum duration of construction activities at the site, with intermittent building phases and occupation durations. This study contributes to understanding the transitional period in the Neolithic era, highlighting changes in settlement dynamics, societal focus, and human impact on the local environment during the late 6th millennium BC. Furthermore, it underscores the significance of high-resolution chronological data in unraveling the complex interplay between humans and their surroundings, particularly in the context of climate influences on past societies.

This research has a significant impact in the fields of archaeology, dendrochronology, and climate science by offering an absolute dating framework for the Dispilio site, shedding light on its construction phases and offering valuable information useful to infer environmental conditions and human adaptations during the studied time frame. This work distinguishes itself from existing literature through its meticulous integration of dendrochronological, radiocarbon, and archaeological analyses, providing a nuanced understanding of societal changes, human-land interactions, and the regional environment during the later Middle Neolithic and Late Neolithic periods.

The work robustly supports the conclusions and claims made, providing detailed and meticulously analyzed data from dendrochronological and radiocarbon studies. The interdisciplinary approach enhances the reliability of the findings, reinforcing the chronological framework and the construction phases of the Dispilio site. The comprehensive analysis of wood samples, combined with high-resolution radiocarbon dating, bolsters the credibility of the interpretations about settlement duration and the societal changes during the Neolithic period.

While the data analysis and interpretation are clear and robust, some limitations can be identified, including the inability to definitively identify certain species due to anatomical similarities or the challenges in precisely determining the dating of certain building phases. These factors have been in any case well addressed in the discussion and sufficiently justified as elements that have no crucial impact on the conclusions of this research and that are open for further investigation in the future.

The methodology employed in this study appears to be rigorous and adheres to the expected standards in the field. Proper sampling and documentation protocols were followed, and tree-ring measurements were conducted using established dendrochronological procedures. The wood taxonomy determination was performed carefully, and the radiocarbon measurements were conducted using a detailed and well-documented approach. Overall, the methodology seems sound and in line with the standard practices of the field.

At the same time materials and methods section appears to provide sufficient detail for the work to be reproduced. Detailed descriptions of the wood sampling process, preparation procedures, and dendrochronological measurements were included. The methodology for radiocarbon measurement and matching was also thoroughly outlined, along with specific details regarding the sample preparation and analytical techniques used. Therefore, researchers in the field should have the necessary information to replicate the study and verify its results.

I thoroughly enjoyed reading this well-accomplished research. I recommend sending the manuscript to an English language reviser to correct any writing mistakes, which I am unable to identify as a non-native English speaker. Additionally, I suggest double-checking the text for typos, as a repeated phrase was noticed in line 363. After addressing these minor corrections, I am delighted to recommend this paper for publication.

Reviewer #2:

Remarks to the Author:

The manuscript 'Absolutely dating the European Neolithic through a rapid 14C excursion' presents new and exciting results from the intersection between archaeology, 14C dating and dendrochronology, with significant relevance for a wider range of fields. The material, methods and results are suitably presented and meet current standards within the fields, at times even exceeding it and creating higher standards for future studies. The results are discussed within a larger framework and in support of the conclusions. With some minor revisions, I find the manuscript well-suited for publication in Nature Communications.

Underneath I include a list of minor points whereupon the manuscript may be further improved.

Overall, the language is ok, but could be improved in several places.

- Line 28-30: "Dispilio is thus the first prehistoric site absolutely dated through a 14C signature (Miyake event), but also the first absolutely, calendar-year dated prehistoric site in the wider Mediterranean region".

i) This depends on the definition of 'prehistory', which is highly regionally dependent. I recommend referencing other studies that have identified Miyake events in archaeological materials, e.g., Kuitens, M., Wallace, B. L., Lindsay, C., Scifo, A., Doeve, P., Jenkins, K., Lindauer, S., Erdil, P., Ledger, P. M., Forbes, V., Vermeeren, C., Friedrich, R., & Dee, M. W. (2022). Evidence for European presence in the Americas in ad 1021. *Nature*, 601(7893), 388-391. <https://doi.org/10.1038/s41586-021-03972-8>
Meadows, J., Zunde, M., Lēģere, L., Dee, M. W., & Hamann, C. (2023). SINGLE-YEAR 14C DATING OF THE LAKE-FORTRESS AT ĀRAIŠI, LATVIA. *Radiocarbon*, 1-11. <https://doi.org/10.1017/RDC.2023.24>
Philippson, B., Fèveille, C., Olsen, J., & Sindbæk, S. M. (2022). Single-year radiocarbon dating anchors Viking Age trade cycles in time. *Nature (London)*, 601(7893), 392-396. <https://doi.org/10.1038/s41586-021-04240-5>

ii) The latter part of the sentence gives the impression that Dispilio is the only prehistoric site that has been absolutely dated, I assume 'absolutely dated to a single-year calendar' is what is meant.

- Introduction: The introduction could benefit from being more archaeologically focussed to better present the context for the scientific results. The relevance of Dispilio in relation to the Mesolithic-Neolithic transition in this region is lacking, which renders the introduction of little relevance to the remaining manuscript. I suggest Shennan (2018) might be of relevance here:

Shennan, S. (2018). *The First Farmers of Europe: An Evolutionary Perspective*. Cambridge University Press. <https://doi.org/10.1017/9781108386029>

- Line 51: "Until now" gives the false impression this is the first study to apply a combined approach of dendrochronology and 14C dating of Miyake events. Please provide references to earlier applications.

- Line 73: Logically the Mesolithic should be mentioned before the Neolithic.

- Line 89: What qualifies Dispilio as a 'premier prehistoric wetland site'? I miss a definition of the type, was it a settlement/fortified site/burial site/etc? Also, was it a commonly occurring type of site for this period and region?

- Line 100: I find it confusing that the text refers to 1700 complete ceramic vessels, but Fig 2 illustrates a fragmented vessel.

- Line 101-02: Using the term network invokes notions of trading and contacts, at least in the minds of archaeologists, but I suspect the authors refer to artefact typology here.

- Line 105-06: I suggest describing these as 'floating chronologies'.

- Line 110: see earlier comment on line 28-30.

- Line 130: Fig 4 is introduced before Fig 3.

- Figure 4: Please state clearly what methods were used to anchor the floating tree-ring-series chronologically, i.e. was it the "conventional radiocarbon wiggle-matching models" (line 149), or perhaps the use of the Miyake event?

- Line 147, Supplementary S1: provides a wiggle match model of oak block ID Disp-6001, but 1) the purpose of the model is unclear, and 2) block Disp-6011 and related 14C dates are neither described

nor included elsewhere in the manuscript or supplementary material.

- Line 149ff: I find it confusing throughout the paper what wooden block were analysed, their species, ID, number of tree rings and which rings that have been 14C dated. I suggest inserting a table with the information. This might also clarify how you can wiggle match "blocks of 1-11 tree-rings"?
- Figure 3: The box in the topmost right corner of figure 3a is not described in the figure text.
- Line 223: Please briefly state what is meant here by the term 'cross dated', i.e. dendrodated wooden objects that has been cross dated to the absolutely dated master curve.
- Line 244: How can the timeframe be 'universal' if it is regional?
- Line 246: Please state what makes "the absolute dating and duration" unique. The precision? Is the duration particularly short/long?
- Line 290: Why are these other sites described as 'peripheral'? In relation to their regional significance or simply due to their geographical position in relation to Dispilio?
- Line 291: See earlier comment about the use of the term 'network', I assume the authors refer to 'chronological framework'.
- Line 310: Annual radiocarbon measurements (missing word).
- Line 311: Improving the resolution of the radiocarbon calibration curve, thereby improving the precision of the calibrations.
- Line 354-58: Please state that this is based on modern data.

Reviewer #3:

Remarks to the Author:

The paper is interesting and the approach is new. However, I think that the introduction should be re-written, some of the references updated and others changed. There is quite a bit of confusion regarding the Mediterranean or Balkan location of the site that should be clarified. The archaeological cultural aspect of the site is never mentioned. This should be added somewhere. In my opinion the chronological method employed is interesting and innovative. But one of the points regards also the settlement duration period. There are many points to be clarified and revised. This is my general impression

1 **Response to the reviewers' comments**

3
4 We would like to thank all the reviewers for the kind and thorough reviews which improve the clarity
5 of the paper. Reviewers' comments are in original with plain text, and the authors' responses are
6 below each comment in a rectangular box. Reviewer #3 comments, which were originally
7 incorporated into a PDF of the manuscript, were copied and pasted here.

8
9 **Reviewer comments and responses**

10
11 **1. Reviewer #1 (Remarks to the Author):**

12
13 This study at Dispilio revealed robust oak and juniper tree-ring chronologies, providing a 120-year
14 oak TRW chronology and a 303-year juniper TRW chronology. The detection of the 5259 BC Miyake
15 event in the juniper chronology allowed for the absolute dating of both chronologies, placing the
16 end-dates at 5140 BC and 5311 BC, respectively. Additionally, the findings suggested a 188-year
17 minimum duration of construction activities at the site, with intermittent building phases and
18 occupation durations. This study contributes to understanding the transitional period in the
19 Neolithic era, highlighting changes in settlement dynamics, societal focus, and human impact on
20 the local environment during the late 6th millennium BC. Furthermore, it underscores the
21 significance of high-resolution chronological data in unraveling the complex interplay between
22 humans and their surroundings, particularly in the context of climate influences on past societies.

23
24 This research has a significant impact in the fields of archaeology, dendrochronology, and climate
25 science by offering an absolute dating framework for the Dispilio site, shedding light on its
26 construction phases and offering valuable information useful to infer environmental conditions and
27 human adaptations during the studied time frame. This work distinguishes itself from existing
28 literature through its meticulous integration of dendrochronological, radiocarbon, and
29 archaeological analyses, providing a nuanced understanding of societal changes, human-land
30 interactions, and the regional environment during the later Middle Neolithic and Late Neolithic
31 periods.

32
33 The work robustly supports the conclusions and claims made, providing detailed and meticulously
34 analyzed data from dendrochronological and radiocarbon studies. The interdisciplinary approach
35 enhances the reliability of the findings, reinforcing the chronological framework and the
36 construction phases of the Dispilio site. The comprehensive analysis of wood samples, combined
37 with high-resolution radiocarbon dating, bolsters the credibility of the interpretations about
38 settlement duration and the societal changes during the Neolithic period.

39
40 While the data analysis and interpretation are clear and robust, some limitations can be identified,
41 including the inability to definitively identify certain species due to anatomical similarities or the
42 challenges in precisely determining the dating of certain building phases. These factors have been
43 in any case well addressed in the discussion and sufficiently justified as elements that have no

44 crucial impact on the conclusions of this research and that are open for further investigation in the
45 future.

46
47 The methodology employed in this study appears to be rigorous and adheres to the expected
48 standards in the field. Proper sampling and documentation protocols were followed, and tree-ring
49 measurements were conducted using established dendrochronological procedures. The wood
50 taxonomy determination was performed carefully, and the radiocarbon measurements were
51 conducted using a detailed and well-documented approach. Overall, the methodology seems
52 sound and in line with the standard practices of the field.

53 At the same time materials and methods section appears to provide sufficient detail for the work to
54 be reproduced. Detailed descriptions of the wood sampling process, preparation procedures, and
55 dendrochronological measurements were included. The methodology for radiocarbon
56 measurement and matching was also thoroughly outlined, along with specific details regarding the
57 sample preparation and analytical techniques used. Therefore, researchers in the field should have
58 the necessary information to replicate the study and verify its results.

59
60 I thoroughly enjoyed reading this well-accomplished research. I recommend sending the
61 manuscript to an English language reviser to correct any writing mistakes, which I am unable to
62 identify as a non-native English speaker. Additionally, I suggest double-checking the text for typos,
63 as a repeated phrase was noticed in line 363. After addressing these minor corrections, I am
64 delighted to recommend this paper for publication.

We would like to thank Reviewer #1 for the thorough review and recommendation! The manuscript has been revised by a native English speaker who is a professional in scientific writing.

Also, the mistake on line 363 was corrected accordingly

65
66
67

68

69 **2. Reviewer #2 (Remarks to the Author):**

70

71 The manuscript 'Absolutely dating the European Neolithic through a rapid 14C excursion' presents
72 new and exiting results from the intersection between archaeology, 14C dating and
73 dendrochronology, with significant relevance for a wider range of fields. The material, methods and
74 results are suitably presented and meet current standards within the fields, at times even exceeding
75 it and creating higher standards for future studies. The results are discussed within a larger
76 framework and in support of the conclusions. With some minor revisions, I find the manuscript well-
77 suited for publication in Nature Communications.

78 Underneath I include a list of minor points whereupon the manuscript may be further improved.

79 Overall, the language is ok, but could be improved in several places.

80 • Line 28-30: "Dispilio is thus the first prehistoric site absolutely dated through a 14C signature
81 (Miyake event), but also the first absolutely, calendar-year dated prehistoric site in the wider
82 Mediterranean region".

83 i) This depends on the definition of 'prehistory', which is highly regionally dependent. I recommend
84 referencing other studies that have identified Miyake events in archaeological materials, e.g.,

85 Kuitens, M., Wallace, B. L., Lindsay, C., Scifo, A., Doeve, P., Jenkins, K., Lindauer, S., Erdil, P., Ledger,
86 P. M., Forbes, V., Vermeeren, C., Friedrich, R., & Dee, M. W. (2022). Evidence for European presence
87 in the Americas in ad 1021. *Nature*, 601(7893), 388-391. [https://doi.org/10.1038/s41586-021-03972-](https://doi.org/10.1038/s41586-021-03972-8)
88 [8](https://doi.org/10.1038/s41586-021-03972-8)
89 Meadows, J., Zunde, M., Lēgere, L., Dee, M. W., & Hamann, C. (2023). SINGLE-YEAR 14C DATING OF
90 THE LAKE-FORTRESS AT ĀRAIŠI, LATVIA. *Radiocarbon*, 1-11. <https://doi.org/10.1017/RDC.2023.24>
91 Philippsen, B., Feveile, C., Olsen, J., & Sindbæk, S. M. (2022). Single-year radiocarbon dating
92 anchors Viking Age trade cycles in time. *Nature (London)*, 601(7893), 392-396.
93 <https://doi.org/10.1038/s41586-021-04240-5>
94 ii) The latter part of the sentence gives the impression that Dispilio is the only prehistoric site that
95 has been absolutely dated, I assume ‘absolutely dated to a single-year calendar’ is what is meant.

We would like to thank Reviewer #2 for the very thorough and kind review!
The suggested changes were fully incorporated, and the abstract was clarified accordingly by
specifying that what is meant is European prehistory and calendar-year absolute dating.

96
97 • Introduction: The introduction could benefit from being more archaeologically focussed to better
98 present the context for the scientific results. The relevance of Dispilio in relation to the Mesolithic-
99 Neolithic transition in this region is lacking, which renders the introduction of little relevance to the
100 remaining manuscript. I suggest Shennan (2018) might be of relevance here:
101 Shennan, S. (2018). *The First Farmers of Europe: An Evolutionary Perspective*. Cambridge University
102 Press. <https://doi.org/10.1017/9781108386029>

At the beginning of the introduction, we provide a brief, general temporal and
phenomenological setting of the Neolithic as it is defined in western Eurasia. We clarified the
spatial framework within which this study is set. The paragraphs that follow provide a
comprehensive overview of the methodology combining dendrochronological, archaeological,
and 14C information. Further, the reader is introduced to wetland archaeological sites that are
exceptionally significant in prehistoric archaeology considering the often-unique preservation of
organic material, including wood. Lastly, detailed information is provided on the setting of the
site of Dispilio, its chronology, regional significance, and wooden artifacts. We clarified
potentially misleading parts, as the article does not deal with the Mesolithic–Neolithic transition,
which would have occurred over a millennium before the absolutely dated phases at Dispilio. The
Meso-Neolithic transition itself is a topic with very little evidence considering the scarcity of
Mesolithic sites in the Balkans.

103
104 • Line 51: “Until now” gives the false impression this is the first study to apply a combined approach
105 of dendrochronology and 14C dating of Miyake events. 4.

This was clarified by using instead the expression “until recently”, as the use of Miyake
events for dating historical or archaeological sites was first used only in 2014 (Wacker et al.,
2014, reference added), and since then only about a dozen studies have appeared using Miyake
events. We also addressed this by expanding references to other studies who have conducted this
sort of work.

106

107 • Line 73: Logically the Mesolithic should be mentioned before the Neolithic.

Corrected.

108

109

110

111 • Line 89: What qualifies Dispilio as a ‘premier prehistoric wetland site’? I miss a definition of the
112 type, was it a settlement/fortified site/burial site/etc? Also, was it a commonly occurring type of site
113 for this period and region?

We clarified this. Dispilio’s importance lies in the fact that it is the only wetland settlement or ‘pile-dwelling’ in the Balkans to be systematically excavated over multiple years and on a large area.

114

115 • Line 100: I find it confusing that the text refers to 1700 complete ceramic vessels, but Fig 2
116 illustrates a fragmented vessel.

Sentence clarified. Under “complete” in European Neolithic sites, one can include vessels from which most parts are preserved in such a manner that even the missing pieces can be reconstructed with great certainty. For instance, intentional, perhaps ritual, vessel breaking is a common feature observed in various sites during the Neolithic in Southeastern Europe, a practice which leads to the formation of sites with great amounts of pottery that do not have any intact “complete” vessels, but have many vessels whose parts are all preserved in the layer. Dispilio is characterised by its large collection of both in situ complete and also fully restored vessels. The vessel in Figure 2, considering its bilateral symmetry, is one such vessel for which all parts necessary for full restoration have been preserved, but the full restoration has not yet taken place, so only the original parts are visible. Nevertheless, the reference to this figure is removed to avoid unintentional confusion. We thank the reviewer for raising this.

117

118 • Line 101-02: Using the term network invokes notions of trading and contacts, at least in the minds
119 of archaeologists, but I suspect the authors refer to artefact typology here.

Thank you for this observation, we clarified that what is referred to as ‘network’ is a ceramics typology network.

120

121 • Line 105-06: I suggest describing these as ‘floating chronologies’.

It is clarified that we refer to floating tree-ring width chronologies. The oak and juniper tree-ring chronologies are then described in detail in the Results section of the article and are now absolutely dated.

122

123 • Line 110: see earlier comment on line 28-30.

Thank you for the suggestion, definition clarified.

124

125 • Line 130: Fig 4 is introduced before Fig 3.

Corrected.

126

127 • Figure 4: Please state clearly what methods were used to anchor the floating tree-ring-series
128 chronologically, i.e. was it the “conventional radiocarbon wiggle-matching models” (line 149), or
129 perhaps the use of the Miyake event?

Figure 4 caption clarified and expanded as suggested.

130

131 • Line 147, Supplementary S1: provides a wiggle match model of oak block ID Disp-6001, but 1) the
132 purpose of the model is unclear, and 2) block Disp-6011 and related 14C dates are neither
133 described nor included elsewhere in the manuscript or supplementary material.

The wiggle-matching model of the oak chronology is now better defined. 6001 is the ID of the oak chronology. All radiocarbon dates used in the model are included in the OxCal CQL code in the Supplementary Material S1. Table S1.3 in Supplementary Material S1 was added: it lists and describes all the radiocarbon dates included in the wiggle-matching model of the oak chronology. The wiggle-matching of radiocarbon dates from the oak chronology is supporting the dendrochronological cross-dating between the oak and juniper tree-ring chronologies.

134

135 • Line 149ff: I find it confusing throughout the paper what wooden block were analysed, their
136 species, ID, number of tree rings and which rings that have been 14C dated. I suggest inserting a
137 table with the information. This might also clarify how you can wiggle match “blocks of 1-11 tree-
138 rings”?

We agree that “blocks of 1-11 tree-rings” was confusing and have removed this from the text. We have also tried to clarify the associated text which may have added to the confusion. The excavation IDs of the wood samples that were sampled for annual 14C measurements are DISP-10063, DISP-10070, DISP-10206 and DISP-10611. All four wooden samples, cross-sections cut from vertical wooden piles, are described in the Results section of the main article text, and schematically represented in Figure 4, while the photographs of each wood sample are available in Supplementary Material S2. In response to the table suggestion, we actually had Supplementary Table T1 already, which contains some of the relevant information (wood samples’ IDs, rings sampled and corresponding radiocarbon measurements, archaeological site location, 14C lab at which the measurements were performed) but we have added information on the total ring number of each sample. Total ring number and ring-widths measurements were also provided in Supplementary Material S3.3. All juniper 14C measurements discussed in the main article text are annual, i.e. performed on single tree-rings as listed in Supplementary Table T1. However, initially, in order to place the juniper chronology on an absolute time frame, so even before the publication of the Brehm et al.’s (2022) findings, 11 radiocarbon measurements were taken from the juniper chronology to understand its approximate placement on the calendar scale (we now added also these initial juniper radiocarbon measurements to Sup. Mat S1). These initial results were the guide for the identification of the area of the juniper chronology where the Miyake event would be located.

The juniper and oak tree-ring chronologies cross-date dendrochronologically against each other, with an overlap over 108 years, placing the last ring of the oak chronology at 5311 BC, i.e. before the 5259 BC Miyake event. This dendrochronological cross-dating between the two chronologies is also supported by a wiggle-matching model of ^{14}C measurements obtained from blocks of several tree-rings from oak wood samples that make up the oak chronology, and the median ring of each block of tree-rings was used in defining the gaps in the wiggle-matching model of the oak chronology, as described in the updated Supplementary Material S1 Table S1.1. The oak wiggle-matching model results have been described in an earlier article in an edited volume (*Prehistoric Wetland Sites of Southern Europe: Archaeology, Dendrochronology, Palaeoecology and Bioarchaeology*), originally scheduled for publication in 2022, which is already typeset and should be published anytime now (main article ref. Maczkowski et al., 2024).

139

- 140
- Figure 3: The box in the topmost right corner of figure 3a is not described in the figure text.

Yes, we overlooked describing the inset in panel a. Now corrected, rest of Figure 3 caption has also been expanded and clarified, thank you!

141

- 142
- Line 223: Please briefly state what is meant here by the term ‘cross dated’, i.e. dendrodated
- 143 wooden objects that has been cross dated to the absolutely dated master curve.

Paragraph updated accordingly.

144

- 145
- Line 244: How can the timeframe be ‘universal’ if it is regional?

What is meant is an “unanimously agreed upon absolute timeframe”. Sentence clarified.

146

- 147
- Line 246: Please state what makes “the absolute dating and duration” unique. The precision? Is
- 148 the duration particularly short/long?

Yes, it is the precision that makes the absolute dating unique. We further clarified in the paragraph that all estimates of house and settlement occupation durations in south-eastern Europe are based solely on ^{14}C modelling, which has an associated uncertainty that at times can be of several decades, so longer than a generation. Additionally, the geographically closest accurate radiocarbon models of house and settlement durations come from sites in northern Serbia and Hungary, and detailed radiocarbon dating programs focusing on Neolithic settlement durations further south in the rest of the Balkans, but also Anatolia, do not exist.

149

- 150
- Line 290: Why are these other sites described as ‘peripheral’? In relation to their regional
- 151 significance or simply due to their geographical position in relation to Dispilio?

Thank you for this remark, the chosen term is indeed inappropriate. What are meant are the ‘neighbouring’ sites surrounding Dispilio within a relatively short distance, in dendrochronological terms, of ca. 90 km.

152

153 • Line 291: See earlier comment about the use of the term ‘network’, I assume the authors refer to
154 ‘chronological framework’.

Thank you for this remark, the sentence has been clarified. Yes, we refer to the “dendrochronological network”, a term commonly used in dendrochronology, i.e. “tree-ring network”, a term that denotes a geographical network of different tree-ring chronologies which have high correlation values between them, indicating a common regional or synoptic scale climatic signal preserved in the tree-rings (see: ucar.edu). In this sense, the three-centuries long Dispilio juniper chronology can be extended through cross-dating with other wood samples or chronologies from the neighbouring Neolithic sites.

155

156 • Line 310: Annual radiocarbon measurements (missing word).

Missing word inserted, thank you.

157

158 • Line 311: Improving the resolution of the radiocarbon calibration curve, thereby improving the
159 precision of the calibrations.

Sentence clarified accordingly, thank you.

160

161 • Line 354-58: Please state that this is based on modern data.

Clarified that what are described are the modern climatic conditions.

162

163

164

165

166 **3. Reviewer #3 (Remarks to the Author):**

167

168 The paper is interesting and the approach is new. However, I think that the introduction should be
169 re-written, some of the references updated and others changed. There is quite a bit of confusion
170 regarding the Mediterranean or Balkan location of the site that should be clarified. The
171 archaeological cultural aspect of the site is never mentioned. This should be added somewhere. In
172 my opinion the chronological method employed is interesting and innovative. But one of the points
173 regards also the settlement duration period. There are many points to be clarified and revised. This
174 is my general impression

We would like to thank Reviewer #3 for the kind review! The comments of Reviewer #3 were incorporated in the PDF version of the manuscript, so they are copied and pasted here and answered accordingly.

175

176 - Line 39: I would not confuse what happened in the Levant and what happened in Greece for
177 any reason ever because the processes aren't comparable. Moreover, according to the
178 available data, we know almost nothing of the second.

Thank you for the remark! We acknowledge that this might be confusing to the reader. We adapted the sentence accordingly, removing the reference to the Levant.

179

180 - Line 40: The list of Mavropigi dates in the paper by Kozłowski et al etc.. is incomplete. The
181 complete list has been published by E Starnini in EP 2018. See the radiocarbon dates table.
182 The earliest dates along the Aegean coast and those from Western Macedonia are very
183 similar. This is important to stress

Thank you; the suggested references were included. Also, it is stated in the first paragraph of the introduction that the appearance of the Neolithic occurs at approximately at similar times in the Aegean and Northern Greece.

184

185 - Line 41: What do we know of the Late Mesolithic period in the Greek peninsula, apart from
186 Cave Franchthi? Almost nothing 40 years ago and almost nothing nowadays

We absolutely agree, there is very little information on the Mesolithic-Neolithic transition in the region given the scarcity of sites which document both periods, as represented only by Franchthi Cave and Cyclops Cave. However, regardless of the archaeological information and material culture, changing land use practices can be inferred from pollen spectra, tree-rings, and other natural proxies.

187

188 - Line 46: Lake dwelling site..... in Western Macedonia (Greece)

Thank you for the remark. We defined the type of site accordingly. Yes, the site is located in the Greek administrative unit of Western Macedonia, which is bordered to the east by the administrative unit of Central Macedonia, to the north by the country of North Macedonia, to the west by the country of Albania and the Greek administrative unit of Epirus. Given the broader audience at which the journal Nature Communications is aimed at, we use more general geographical terms that are more familiar to a broader audience, rather than terms that include local administrative divisions. The precise location of the site is also illustrated on Figure 1, panels a. and b.

189

190 - Line 89: north-western (Western Macedonia close to Albania etc.,)

It is clarified that the site is situated in northwestern Greece, and its significance is defined as the only systematically excavated pile-dwelling in the Balkans is defined.

191

192 - Line 110: Is this region part of the southern Balkans or the Mediterranean? I think the first

The coastal and inland areas of many parts of the Balkans are often defined as "Mediterranean" by different sources. Both the definition of "Balkans" and "Mediterranean" are often ambiguous as they can have different meanings depending on what is the basis of the definition, whether it is political borders, geography, culture, vegetation, climate, proximity to the sea, etc. This paragraph

discusses tree-ring chronologies constructed for the wider Eastern Mediterranean region, given that the climate signal preserved in tree-rings is often common for trees on a regional and synoptic scale, i.e. up to 1000 km away and even more. The vegetation in northwestern Greece and in the neighbouring areas of Albania and North Macedonia contain plant communities characteristic of the Mediterranean, and the summer climate is also most often described as sub-Mediterranean. The contacts with the sea of these part of the Balkans, and even further north, have been constant since at least the Neolithic (e.g. spondylus shell bracelets in Dispilio) until today. We further clarify this paragraph by specifying that what is meant is absolute calendar-year dating as, to our knowledge, there is no other Neolithic site in the whole Mediterranean basin that has been absolutely dated to single-alendar-year precision.

193

194 - Line 222: Unclear

Thank you for this remark. We acknowledge that it was not entirely clear what was meant. The paragraph was clarified to explain that plotting of wooden piles that have been dendrodated and have the last ring in the same year revealed blueprints of buildings as shown in Figure 4b.

195

196 - Line 246: I think that this should be specified a bit better. What is the Middle Neolithic like in
197 that region and what the Late Neolithic? This should be specified because it is very unclear
198 to me

The sentences were rewritten, and a reference was added to clarify this part. The defining characters and the timing of the transition from Middle to Late Neolithic in the region has not been agreed upon in an absolute timeframe. Furthermore, the terminology has not been agreed upon even across borders, so for instance the 54th–52nd centuries BC in Albania would fall in the Middle Neolithic, in Aegean terminology it is the Late Neolithic I, and in the more general Balkan terminology this would fall at the transition between the two. For some, the latter part of the 6th millennium might even be attributed even to the early Chalcolithic. The division of Early, Middle, and Late Neolithic was established by Weinberg, and has been in place since 1942. Such a division was very useful and meaningful when radiocarbon was not available, and the excavated sites were few. However, with the great number of excavated sites and more detailed excavation techniques during the course of the 20th and 21st centuries produced a much more diverse and nuanced picture of the changes occurring in the archaeological record. However, some generalisations are possible, for instance concerning the pottery decoration, switching from white-on-red to darker pottery with incised decoration in the Late Neolithic. Sites' locations also change, and the Late Neolithic ones are spread in all environments, while population densities increase, and changes in burial customs are documented. These changes vary between both regions and sites. The definition of the Middle and Late Neolithic is outside the scope of our paper. The absolute dating of Dispilio is in fact a step forward towards dealing with the at times confusing concepts associated with the tripartite periodisation of the Neolithic and their correspondence on wider geographical scales.

199

200 - Line 247: Here again is written Balkans! I agree because Kastoria is a Balkan locality

Thank you for this remark. As discussed above in the reply to Line 110, the terms 'Balkan' and 'Mediterranean' are often ambiguous and depend on the definition and the context. The southern parts of the Balkans, especially when considered in terms of vegetation, but also cultural contacts, are part of the wider Mediterranean.

201

202 - Line 248: This is also unclear to me!

The duration of occupation of Neolithic settlements and individual dwellings in Southeastern Europe has been so far estimated only through radiocarbon modelling. And while the radiocarbon models can provide satisfactory resolution at times, this is only possible on sites that have been excavated on a sufficiently large area with modern, detailed excavation techniques where a great number of radiocarbon dates are available, as in the examples presented by Tasic et al., 2016 and Bayliss and Whittle, 2018. However, such techniques are limited by the associated uncertainty of radiocarbon measurements, the replication, and the shape of the radiocarbon calibration curve. Moreover, radiocarbon dates on wood charcoal are susceptible to the so-called 'old wood effect', by which the radiocarbon-dated charcoal might come from the inner part of an old tree, which can yield dates that are too old by up to a century or more. Consequently, the replicated dendrochronological information from Dispilio provides the duration of the settlement phases and the duration of buildings in calendar years, independently of radiocarbon.

203

204 - Line 249: I think the periods during which the site was constructed, this is my impression

Yes, this refers to the period for which we have secure felling-dates: wood samples with the preserved final ring (or bark). From the earliest secure felling-date to the latest, as identified solely from the dendrodated wood samples in the East Sector, there are at least 188 years. The phrase "at least" indicates that we do not know whether some of the undatable wooden piles come from later or earlier than this, or whether other areas of the site preserve piles that document felling dates outside the 188-years range.

205

206 - Line 258: This is an important point to stress. Is the chronology of the settlement
207 construction the same of the settlement life?

We fully recognise the limitations of archaeological data, as archaeological sites are products of very complex, stochastic formation processes. Because we cannot determine what is missing from the archaeological record, interpretations must be made with caution, and, as in the given paragraph, several interpretation options should be listed as starting hypotheses for a future re-evaluation.

208

209 - Page 11: Can they be reported somewhere in the text or in a note?

Thank you for this comment, we presume that it concerns the models depicted in Figure 5. The data is available in the Supplementary Material 4

210

211 - Line 273: can this be better specified? Which cultural aspects were active in the area during
212 that period?

The changes observed at the end of the 6th millennium BC include rising population numbers, and an associated increase in the number of settlements. As cited references describe, there are changes in architecture and settlement organisation, in pottery decoration, and the first proxy indication of environmental changes most probably induced by anthropic activity.

213

214 - Line 281: What are the subperiods?

Periods clarified as suggested.

215

216 - Line 282: This is not so strange all over the Balkans and other countries of Europe

Yes, this phenomenon is attested in other regions as well, as the Neolithic sedentary farming societies would have chosen locations within reach of water sources.

217

218

219

220

Reviewers' Comments:

Reviewer #2:

Remarks to the Author:

The authors have revised the manuscript in a satisfying manner, and I am content with the result. The only thing I object to that there are no references to previous studies identifying single years in archaeological material, despite my previous suggestion (see extract of my previous comments below). The text now reads: "Dispilio is thus the first European prehistoric Neolithic site dated absolutely dated to a calendar year through a 14 35 C- signature (Miyake event)", which is not wrong, but it does give the implicit message that this is also the first study to date an archaeological site to a single calendar year - which it is not. I strongly recommend referencing and acknowledging previous publications applying comparable methodologies. This will not remove the importance and impact of the present study.

- Line 28-30: "Dispilio is thus the first prehistoric site absolutely dated through a 14C signature (Miyake event), but also the first absolutely, calendar-year dated prehistoric site in the wider Mediterranean region".

Reviewer 2: This depends on the definition of 'prehistory', which is highly regionally dependent. I recommend referencing other studies that have identified Miyake events in archaeological materials, e.g.,

Kuitens, M., Wallace, B. L., Lindsay, C., Scifo, A., Doeve, P., Jenkins, K., Lindauer, S., Erdil, P., Ledger, P. M., Forbes, V., Vermeeren, C., Friedrich, R., & Dee, M. W. (2022). Evidence for European presence in the Americas in ad 1021. *Nature*, 601(7893), 388-391. <https://doi.org/10.1038/s41586-021-03972-8>
Meadows, J., Zunde, M., Lēģere, L., Dee, M. W., & Hamann, C. (2023). SINGLE-YEAR 14C DATING OF THE LAKE-FORTRESS AT ĀRAIŠĪ, LATVIA. *Radiocarbon*, 1-11. <https://doi.org/10.1017/RDC.2023.24>
Philippesen, B., Feveile, C., Olsen, J., & Sindbæk, S. M. (2022). Single-year radiocarbon dating anchors Viking Age trade cycles in time. *Nature (London)*, 601(7893), 392-396. <https://doi.org/10.1038/s41586-021-04240-5>

Reviewer #3:

Remarks to the Author:

The paper has been accurately revised. I have only three more recommendations which may be useful

You use sometimes northern Greece, sometimes northwestern Greece. Maybe better to unify the terminology (northwestern??)

Line 26. Perhaps better Aegean and Balkan.... add also Balkan

Line 53. Would be more correct to add the reference Biagi, Shennan, Spataro... Rapid rivers and slow seas that was the first to address the problem. Easy to find on the internet

Response to the 2nd round of reviewers' comments

We would like to thank all the reviewers for the kind and thorough reviews which improve the clarity of the paper. Reviewers' comments are in original with plain text, and the authors' responses are below each comment in a rectangular box. Second round of review did not include any additional comments from Reviewer #1.

Reviewers' comments and responses:

Reviewer #2 (Remarks to the Author):

The authors have revised the manuscript in a satisfying manner, and I am content with the result. The only thing I object to that there are no references to previous studies identifying single years in archaeological material, despite my previous suggestion (see extract of my previous comments below). The text now reads: "Dispilio is thus the first European prehistoric Neolithic site dated absolutely dated to a calendar year through a 14 35 C- signature (Miyake event)", which is not wrong, but it does give the implicit message that this is also the first study to date an archaeological site to a single calendar year - which it is not. I strongly recommend referencing and acknowledging previous publications applying comparable methodologies. This will not remove the importance and impact of the present study.

- Line 28-30: "Dispilio is thus the first prehistoric site absolutely dated through a 14C signature (Miyake event), but also the first absolutely, calendar-year dated prehistoric site in the wider Mediterranean region".

Reviewer 2: This depends on the definition of 'prehistory', which is highly regionally dependent. I recommend referencing other studies that have identified Miyake events in archaeological materials, e.g.,

Kuitems, M., Wallace, B. L., Lindsay, C., Scifo, A., Doeve, P., Jenkins, K., Lindauer, S., Erdil, P., Ledger, P. M., Forbes, V., Vermeeren, C., Friedrich, R., & Dee, M. W. (2022). Evidence for European presence in the Americas in ad 1021. *Nature*, 601(7893), 388-391. <https://doi.org/10.1038/s41586-021-03972-8>

Meadows, J., Zunde, M., Lēgere, L., Dee, M. W., & Hamann, C. (2023). SINGLE-YEAR 14C DATING OF THE LAKE-FORTRESS AT ĀRAIŠĪ, LATVIA. *Radiocarbon*, 1-11. <https://doi.org/10.1017/RDC.2023.24>

Philippsen, B., Feveile, C., Olsen, J., & Sindbæk, S. M. (2022). Single-year radiocarbon dating anchors Viking Age trade cycles in time. *Nature (London)*, 601(7893), 392-396. <https://doi.org/10.1038/s41586-021-04240-5>

We would again like to thank Reviewer #2 for the very thorough and helpful review in the first and second round! References to Wacker et al. 2014 and Kuitmes et al, 2022 were originally included in the manuscript. We added the other two references suggested by the reviewer (Meadows et al., 2023 and Philippsen et al., 2022).

Reviewer #3 (Remarks to the Author):

The paper has been accurately revised. I have only three more recommendations which may be useful

You use sometimes northern Greece, sometimes northwestern Greece. Maybe better to unify the terminology (northwestern??)

We would also like to thank Reviewer #3 for the review and helpful comments! We adapted the terminology, accordingly, going from the broader 'Northern Greece' to the narrower 'Northwestern' as we go from broader regional setting to the more specific context of the site within the text.

Line 26. Perhaps better Aegean and Balkan.... add also Balkan

Suggestion added.

Line 53. Would be more correct to add the reference Biagi, Shennan, Spataro... Rapid rivers and slow seas that was the first to address the problem. Easy to find on the internet

We could not identify the line for which this reference was suggested. Additionally, the relevance of the suggested reference "New Observations on the Radiocarbon Chronology of the Starcevo-Cris and Koros Cultures; Rapid Rivers...." by Biagi et al., 2005 is limited to the subject of our study, considering that Biagi et al. focus on the Early Neolithic in Romania, Hungary and Northern Serbia.